# CD26/DPP4 as a Therapeutic Target in Nonalcoholic Steatohepatitis Associated Hepatocellular Carcinoma

**DOI:** 10.3390/cancers14020454

**Published:** 2022-01-17

**Authors:** Sohji Nishina, Keisuke Hino

**Affiliations:** Department of Hepatology and Pancreatology, Kawasaki Medical School, 577 Matsushima, Kurashiki, Okayama 701-0192, Japan; f018ep@med.kawasaki-m.ac.jp

**Keywords:** DPP4 inhibitor, immunomodulation, chemokine, eosinophil, macrophage polarization, insulin resistance, obesity

## Abstract

**Simple Summary:**

CD26/DPP4 has been reported to attenuate anticancer immunity via chemokine cleavage and to promote insulin resistance and inflammation in the liver and/or adipose tissue via dysregulation of macrophage M1/M2 polarization. These results suggest the promotive roles of CD26/DPP4 especially in nonalcoholic steatohepatitis (NASH) associated hepatocellular carcinoma (HCC). In this review, we discuss the biological roles of CD26/DPP4 in the development and progression of NASH associated HCC and the potential of DPP4 inhibitors as therapeutic agents for HCC.

**Abstract:**

Hepatocellular carcinoma (HCC) is generally considered an “immune-cold” cancer since T cells are not observed abundantly in HCC tumor tissue. Combination therapy with immune checkpoint inhibitors and vascular endothelial growth factor (VEGF) inhibitors is currently recognized as a first-line systemic treatment for advanced-stage HCC. Immunologically, immune checkpoint inhibitors influence the recognition of cancer cells by T cells, and VEGF inhibitors influence the infiltration of T cells into tumors. However, no drugs that facilitate the trafficking of T cells toward tumors have been developed. Chemokines are promising agents that activate T cell trafficking. On the other hand, metabolic factors such as obesity and insulin resistance are considered risk factors for HCC development. CD26/dipeptidyl peptidase 4 (DPP4) functions as a serine protease, selectively cleaving polypeptides with a proline or alanine at the penultimate N-terminal position, such as chemokines. Recently, CD26/DPP4 has been reported to attenuate anticancer immunity via chemokine cleavage and to promote insulin resistance and inflammation in the liver and/or adipose tissue via dysregulation of macrophage M1/M2 polarization. In this review, we discuss the promotive roles of CD26/DPP4 in HCC development and progression and the potential of DPP4 inhibitors as therapeutic agents for HCC.

## 1. Introduction

Hepatocellular carcinoma (HCC) is the third leading cause of cancer-related death worldwide. Patients with advanced HCC, including those with macrovascular invasion or extrahepatic spread, have a poor prognosis, with a survival time of 6–8 months or 1-year survival rate of 25% [1]. Recently, a combination therapy regimen with an anti-programmed cell death 1-ligand 1 (PD-L1) antibody (atezolizumab), and an anti-vascular endothelial growth factor (VEGF) antibody (bevacizumab) was recommended as the first-line systemic therapy for advanced-stage HCC, as defined by the modified Barcelona Clinic Liver Cancer staging system, at the American Association for the Study of Liver Diseases (AASLD) consensus conference [2,3]. Although this type of immunotherapy, in addition to treatment with multityrosine kinase inhibitors such as sorafenib and lenvatinib has prolonged the overall survival of patients with advanced HCC in the past decade [4,5], their therapeutic effects of these agents are still unsatisfactory. Therefore, it is worth investigating new therapeutic targets or agents that have anti-tumor effects in patients with HCC.

Chemokines are promising agents that activate T cell trafficking. Dipeptidyl peptidase 4 (DPP4) is expressed as both a type II cell surface protein (CD26) and as a soluble protein lacking intracellular and transmembrane domains [6]. DPP4 functions as a serine protease, selectively cleaving proteins at the N-terminal penultimate proline or alanine residue [7,8]. Several chemokines have been shown to be processed in vitro by DPP4 [9], which potentially results in the failure of T cell chemotaxis. These results suggest that inhibition of DPP4 may have the potential to exert anti-tumor effects in HCC by modulating tumor immunity. Additionally, CD26 expression is reported to be upregulated in tumor tissue in patients with HCC [10,11,12]. In this review, we discuss the biological roles of CD26/DPP4 in the development and progression of HCC and consider the potential of CD26/DPP4 as a therapeutic target in HCC.

## 2. Expression of CD26/DPP4 in HCC

CD26 is a membrane glycoprotein widely expressed in various cells, such as T lymphocytes and epithelial and endothelial cells [13]. The CD26 expression levels are altered in various types of cancers. CD26 overexpression is observed in ovarian cancer [14], thyroid cancer [15], malignant mesothelioma [16], colorectal cancer [17], prostate cancer [18], and osteosarcoma [19]. In contrast, CD26 is downregulated in melanoma [17]. While CD26 mostly acts as an oncogene, it acts as a tumor suppressor gene in some cancers.

CD26 expression is localized to the bile canalicular plasma membrane in the normal liver and is recognized as a useful bile canalicular enzyme for assessing the functional polarization of hepatocytes [12]. However, the distribution pattern of CD26 is altered in HCC, which exhibits an increase in CD26 expression to varying degrees [11,12]. CD26 mRNA levels in tumor tissues were found to be significantly increased compared with those in the corresponding adjacent noncancerous lesions, and a higher CD26 expression level was associated with significantly larger tumor size [10]. CD26 knockdown with siRNA led to suppression of tumor growth from HCC cell lines [10]. We also found that HCC tumors with high CD26 expression showed higher alpha-fetoprotein levels, more advanced tumor stages, poorer histologic differentiation, higher tumor cell infiltration into the tumor capsule, and greater cell proliferation than those with lower CD26 expression [11]. In addition, the cumulative recurrence rate of HCC was higher, and the overall survival time was shorter in HCC patients with high CD26 expression [11]. These results suggest that CD26 plays an oncogenic role and serves as a negative prognostic marker in HCC.

## 3. Induction of CD26/DPP4 Expression in Obesity and HCC

How is CD26 expression upregulated in hepatocytes? A recent study demonstrated that obesity triggers a Ca^2+^−Ca^2+^-calmodulin-dependent protein kinase II (CaMKII)−activating transcription factor 4 (ATF4) pathway in hepatocytes, leading to induction of DPP4 expression and secretion of soluble DPP4 (Figure 1A), which acts cooperatively with plasma factor Xa to promote inflammation and insulin resistance [20]. Factor X triggers inflammation in endothelial cells and leukocytes [21,22]. A meta-analysis of cohort studies has found that excess body weight is associated with an increased risk of liver cancer [23]. HCC has recently been linked to non-alcoholic fatty liver disease (NAFLD), a major hepatic manifestation of the metabolic disorder [24]. These results suggest that the increased expression of CD26/DPP4 in HCC may be linked to obesity in some patients.

The serine exopeptidase activity of DPP4 accounts for its biological functions, including hepatic lipogenesis and chemokine regulation [25]. SerpinB3, a serine protease inhibitor, has been reported to be undetectable in normal hepatocytes, while its expression has been found to be progressively increased in patients with chronic liver diseases [26], dysplastic nodules [27], and HCC [28,29]. CD26/DPP4 and SerpinB3 were localized in the same area of HCC tumors and in both parenchymal hepatocytes and proliferating small bile ducts in surrounding cirrhotic tissue [30]. Interestingly, cell lines overexpressing SerpinB3 exhibited upregulation of CD26/DPP4, which likely occurred as a feedback mechanism due to SerpinB3-mediated inhibition of the DPP4 enzymatic activity [30].

Circular RNAs (circRNAs), a novel kind of regulatory RNA characterized by a continuous covalent closed loop structure without a 5′-cap or 3′-poly(A) tail, are considered splicing error byproducts. Aberrant expression of circRNAs has been implicated in the initiation and development of various diseases, including cancers [31]. For instance, circTRIM33-12 acts as a sponge for miRNA-191 to inhibit HCC progression [32]. Recently, MET proto-oncogene receptor tyrosine kinase (*MET*)-derived circRNA (circMET) (hsa_circ_0082002, a 1214 bp circRNA) was found to exhibit the greatest overexpression in HCC tissues compared to non-tumor liver tissues among the circRNAs derived from the chromosome 7q21–7q31 region [33]. Overexpression of circMET was found to promote HCC development independent of MET function by inducing epithelial-to-mesenchymal transition (EMT) and enhancing the immunosuppressive tumor microenvironment [33]. Mechanistically, circMET acted as a sponge for microRNA (miR)-30-5p, which downregulated Snail. Snail upregulated DPP4 expression by interacting with the enhancer element (region between 140,810 and 96,828 base pairs upstream of the DPP4 translation start site) of DPP4 [33] (Figure 1B). Thus, overexpression of circMET has been shown to lead to increased expression of DPP4 in HCC.

## 4. Inhibition of CD26/DPP4 Enzymatic Activity

### 4.1. Immunomodulation

#### 4.1.1. Post-Translational Modification of Chemokines

As mentioned earlier, recent progress in the treatment of advanced HCC has shed light on anti-tumor immunity. Deep single-cell RNA sequencing revealed enrichment and clonal expansion of exhausted CD8^+^ T cells and regulatory T cells (Tregs) in human HCC samples [34]. Immune checkpoint inhibitors influence the recognition of cancer cells by T cells [35,36], and VEGF inhibitors influence the infiltration of T cells into tumors [37,38] and the reduction in myeloid-derived suppressor cells and Tregs [39,40,41] in the cancer-immunity cycle [42]. Effective elicitation of anti-tumor immune responses depends on the infiltration of solid tumors by effective T cells, which are guided by chemokines. However, no drugs that facilitate the trafficking of T cells to tumors have been developed.

Chemokines regulate leukocyte trafficking in healthy tissues and in response to stress, such as infection or tissue damage [43]. The activity of chemokines can be influenced by post-translational modifications [43]. DPP4 can cleave many chemokines and cytokines with a proline or alanine in the penultimate N-terminal position [8,11]. Among chemokines, the inflammatory chemokine C-X-C motif chemokine ligand (CXCL) 10 (CXCL10) is readily truncated in vitro, generating an antagonistic form that can engage its receptor CXC chemokine receptors (CXCR) 3 (CXCR3), but does not induce chemotaxis [44]. Comparison of tumor growth in DPP4^−/−^ mice with that in their heterozygous littermates revealed a significant delay in tumor growth when DPP4 was absent, indicating that DPP4 plays a role in tumorigenesis [8]. In addition, we and others reported that inhibition of DPP4 enzymatic activity enhanced anti-tumor effects by preserving biologically active CXCL10 and increasing lymphocyte trafficking into tumors in C57BL/6 mice subcutaneously injected with B16F10 melanoma cells [8] and in a non-alcoholic steatohepatitis (NASH)-related HCC mouse model (STAM^TM^ mice) [11], respectively (Figure 2A). In these studies, the concentrations of full-length CXCL10 (1–77) and truncated CXCL10 (3–77) were determined, and DPP4 inhibitor treatment almost completely suppressed the truncation of CXCL10 in vitro [8,11]. We also determined the IC_50_ values of the tested DPP4 inhibitors (anagliptin and vildagliptin) by measuring the CXCL10 (1–77) concentrations in response to different concentrations of the DPP4 inhibitors. Notably, the IC_50_ values found in one study [11] were similar to previously reported values [45]. In addition, ex vivo analysis using a cell mobility analysis device showed that DPP4 inhibitor treatment induced a significant increase in the mobility of natural killer (NK) cells and T cells obtained from healthy human adults in the presence of CXCL10 and Huh7 cells expressing CD26 [11]. In agreement with these results, a recent report showed that ARI-4175, a pan inhibitor of the DPP4 enzyme family, enhanced CD8^+^ T cell recruitment and activated intrahepatic inflammasome in a murine model of HCC [46].

Consistent with these results in biochemical experiments and animal models, two prospective clinical trials in healthy donors and in chronic hepatitis C patients revealed that DPP4 inhibitor (sitagliptin) treatment resulted in a significant decrease in the truncated CXCL10 (3–77) concentration and a reciprocal increase in CXCL10 (1–77), with only minimal effects on the total levels of the chemokine [47]. In clinical settings, elevated levels of truncated CXCL10 (3–77) have been demonstrated to be associated with increased DPP4 activity, both being negative predictors for viral clearance in patients with either chronic or acute hepatitis C patients [48,49,50]. Most importantly, a retrospective cohort study demonstrated that HCC tissues from patients who received a DPP4 inhibitor (sitagliptin) showed higher CD8^+^ T cell infiltration, while HCC tissues from patients who did not receive sitagliptin showed significantly lower CD8^+^ T cell infiltration [33]. In the clinical perspectives of DPP4 inhibitors in the prevention and treatment of HCC, a nationwide study in Taiwan demonstrated that DPP4 inhibitors decreased the risk of HCC in patients with chronic hepatitis C and type 2 diabetes mellitus [51]. Regarding the induction of DPP4 via the circMET/miR-30-5p/Snail axis in HCC tissues, implantation of C57/BL/6 mice with circMET-transfected Hep1-6 cells was reported to induce an immunosuppressive tumor microenvironment with significant suppression of tumor-infiltrating CD8^+^ T cells [33]. The mechanisms by which the circMET/miR-30-5p/Snail/DPP4 axis induces an immunosuppressive tumor microenvironment involved a decrease in the CXCL10 (1–77) concentration. Moreover, multivariate analysis identified circMET expression as an independent predictor of poor overall survival and high postoperative recurrence in 209 HCC patients [33].

Obesity is a major risk factor for HCC and is typically accompanied by increased levels of serum DPP4 [52]. In a carcinogen-induced rat model of HCC, high-fat diet-induced DPP4 activity facilitated angiogenesis and promoted HCC proliferation and metastasis [53]. Serum DPP4 promoted high-fat diet-mediated angiogenesis through chemokine ligand (CCL) 2 upregulation. However, supplementation with recombinant DPP4 protein failed to promote cancer cell proliferation in vitro, suggesting that DPP4 did not directly affect cancer cell biology in vitro. The DPP4 inhibitor, vildagliptin prevented tumor progression by suppressing the proangiogenic role of CCL2 (Figure 2A) but did not affect VEGF, VEGF receptor 2 or angiopoietin-1 levels. Notably, concomitant changes in serum DPP4 and CCL2 were observed in 210 patients with HCC, and high serum DPP4 activity was associated with poor clinical prognosis [53].

Considering the potential role of DPP4 inhibitors as additional therapeutic agents in current HCC treatment, it should also be noted that the combination of a DPP4 inhibitor (sitagliptin) and an anti-PD1 antibody improved anti-tumor immunity in immunocompetent mice [33]. We recently demonstrated that 2-deoxy-D-glucose (2-DG)-encapsulated poly[lactic-co-glycolic acid] (PLGA) nanoparticles (2-DG-PLGA-NPs) augmented chemokine (CXCL9/CXCL10) production in liver tumors via the interferon-γ-Janus kinase-signal transducer and activator of transcription pathway and 5′ adenosine monophosphate-activated protein kinase-mediated suppression of histone H3 lysine 27 trimethylation (H3K27me3) [54]. Although this type of chemokine induction is totally different from that induced by DPP4 inhibitors, 2-DG-PLGA-NPs also enhance CD8^+^ T cell infiltration into liver tumors, and not only amplified the anti-tumor effects induced by the anti-PD1 antibody but also suppressed anti-PD1-resistant tumors [54]. Steatohepatitis is reported to reduce the abilities of immunotherapeutic agents to inhibit liver tumor growth by reducing tumor infiltration by CD4^+^ T cells and effector memory cells [55,56,57]. In addition, NASH has been shown to limit anti-tumor surveillance in immunotherapy-treated HCC [58]. These results emphasize the importance of chemokine induction in enhancing the therapeutic effects of currently used immunotherapies for HCC.

Thus, DPP4 inhibitors are generally thought to have suppressive effects on hepatocarcinogenesis and/or HCC progression. However, DPP4 inhibitors may promote cancer progression through inhibition of chemokine posttranscriptional modification if the targeted chemokines enhance cancer progression. CXCL12 is a substrate of DPP4 [59] and is inactivated by exopeptidases, such as DPP4, matrix metalloproteinase (MMP)-2, and MMP9 [60]. CXCL12 binds to CXCR4 and CXCR7 and subsequently regulates tumor growth and metastasis [61]. The CXCL12/CXCR4 axis plays a critical role in directing the metastasis of CXCR4^+^ cancer cells to organs that express high CXCL12 levels in breast cancer [62]. In this respect, inhibition of DPP4 has been reported to accelerate EMT and breast cancer metastasis via CXCL12/CXCR4/mammalian target of rapamycin (mTOR) signaling [63]. In contrast, a subpopulation of CD26^+^ cells is reported to be expressed in both primary and metastatic tumors in colorectal cancer patients with liver metastasis, suggesting that CD26^+^ cancer cells are associated with enhanced invasiveness and chemoresistance [64]. Indeed, the DPP4 inhibitor vildagliptin has been demonstrated to suppress lung metastasis of colorectal cancer via reduced autophagy, increased apoptosis, and inhibition of the cell cycle regulator phospho-CDC2 (pCDC2) in a syngeneic mouse model [65]. Thus, these results suggest that the effects of DPP4 inhibitors on the metastasis of cancer cells may be dependent on cancer cell type or experimental model.

#### 4.1.2. Eosinophil Migration

Although eosinophils play an important role in eliminating parasites, they have also been reported to participate in the control of tumor growth [66,67,68]. In addition, eosinophils might support anti-tumor immune responses indirectly by facilitating T cell migration into tumors [66]. Recently, Hollande et al. reported a T cell-independent anti-tumor immune response elicited by eosinophils in syngeneic mouse models of HCC and breast cancer [69]. Notably, administration of a DPP4 inhibitor resulted in higher concentrations of the chemokines CCL11, interleukin (IL)-4, IL-5, and IL-33 in tumor extracts and increased migration of eosinophils into solid tumors. The enhanced anti-tumor immune response was maintained in mice lacking lymphocytes but was ablated after depletion of eosinophils or treatment with degranulation inhibitors. Additionally, this group demonstrated that tumor cell expression of the alarmin IL-33 was necessary and sufficient for eosinophil-mediated anti-tumor responses and that this mechanism enhanced the efficacy of immune checkpoint inhibitors. Mechanistically, expression of the alarmin IL-33 in tumors was a key inducer of CCL11 production, with DPP4 inhibitor treatment protecting the chemokine response and enhancing eosinophil migration since it has been demonstrated that CCL11 functions as an eosinophil chemoattractant and that IL-4, IL-5, and IL-33 are cytokines associated with eosinophil activation and effector functions [70,71] (Figure 2B). Thus, IL-33 has dual functions in regulating eosinophil-dependent anti-tumor immunity by upregulating CCL11 expression and stimulating eosinophil degranulation [72]. However, the IL-33-responsive cellular source of CCL11 is currently unknown. Although eosinophils have been reported to improve vascular healing [66], no increase in blood vessel number was found in tumors harvested from mice treated with a DPP4 inhibitor. In this eosinophil-mediated anti-tumor response, CCL11, an eosinophil-selective chemokine in the ‘eotaxin subfamily,’ was demonstrated to be cleaved by DPP4 into a truncated form (CCL11_3–74_) with diminished chemoattractant properties [69], as found for CXCL10 cleavage by DPP4.

### 4.2. Metabolic Modulation

#### 4.2.1. Inhibition of Inflammation in Liver and/or Adipose Tissue

The tumorigenic effects of insulin resistance and complementary hyperinsulinemia are directly mediated by insulin signaling or indirectly related to changes in the metabolism of endogenous hormones, such as insulin-like growth factor I. Conversely, insulin resistance may be a consequence of obesity and hepatic inflammation, both of which promote tumorigenesis [73]. Thus, insulin resistance is closely related to hepatocarcinogenesis. Hepatocyte DPP4 expression in obesity has been demonstrated to increase the level of soluble DPP4 in serum and promote visceral adipose tissue (VAT) inflammation and insulin resistance, suggesting that crosstalk between the liver and VAT can exacerbate metabolic disorder in the context of obesity [20]. Mechanistically, soluble DPP4 acts cooperatively with plasma factor Xa, and these two proteins activate two separate upstream pathways (the caveolin-1 [CAV1]-interleukin-1 receptor-associated kinase [IRAK1]-TGF-β activated kinase 1 [TAK1] pathway for DPP4 and the protease-activated receptor 2 [PAR2]-RAF1 pathway for factor Xa) that synergistically stimulate extracellular signal-regulated kinase (ERK)1/2 and nuclear factor kappa-light-chain-enhancer of activated B cells (NF-κB) to induce monocyte chemokine protein 1 (MCP1) and IL-6 expression in adipose tissue macrophages [20] (Figure 3). Hepatocyte DPP4 expression is considered a risk factor for HCC development and, therefore, could be a potential therapeutic target. However, notably, silencing DPP4 expression in hepatocytes suppressed inflammation of VAT and insulin resistance, but DPP4 inhibition by oral inhibitors did not [20]. These results may be explained by the hypothesis that inhibition of circulating DPP4 activity does not fully block VAT inflammation due to some other action of DPP4 inhibitors, probably related to an increase in plasma insulin [74].

DPP4 was also found to be an adipokine by comprehensive proteomic profiling of the human adipocyte secretome [52] (Figure 3). In this study, serum DPP4 concentrations were significantly higher in obese subjects than in lean subjects, and DPP4 release was strongly correlated with adipocyte size, suggesting that adipocytes are an important source of DPP4 in obesity. Importantly, DPP4 released from adipose tissue was shown to correlate positively with an increased risk score for metabolic syndrome [52]. The risk of HCC development in patients with noncirrhotic liver disease has been demonstrated to be increased 5.4-fold in patients with NAFLD and 5-fold in patients with metabolic syndrome compared to patients with hepatitis C virus infection [75]. Thus, DPP4 secretion from adipocytes is also a risk factor for hepatocarcinogenesis.

In addition to the role of DPP4 as an adipokine with the potential to induce insulin resistance, DPP4 expressed on dendric cells/macrophages play a critical role in potentiating inflammation of adipose tissue in obesity [76]. DPP4 has been reported to be predominantly expressed in M1-polarized macrophages compared with adipocytes in white adipose tissue of diet-induced obese mice [76,77]. The frequency of DPP4^+^ adipose tissue macrophages was higher in obese mice than in lean mice [77], suggesting that DPP4^+^ macrophages are a major source of circulating DPP4 in the obese state. Importantly, the DPP4 inhibitor linagliptin induced a reciprocal decrease in M1 macrophages and an increase in M2 macrophages in white adipose tissue and liver tissue of mice, resulting in attenuation of obesity-induced inflammation and insulin resistance [77]. (Figure 3) Macrophage inflammatory protein (MIP)-1α becomes the most efficient monocyte chemoattractant after cleavage by DPP4 [78,79]. Considering that macrophage accumulation in white adipose tissue and liver tissue was ameliorated with polarization toward an anti-inflammatory phenotype in MIP-1α^−/−^ mice [77], MIP-1α may be a substrate of DPP4 for the regulation of macrophage polarization in the obese state.

Targeting macrophage activation with the DPP4 inhibitor anagliptin has also been performed in genetically obese melanocortin 4 receptor-deficient (MC4R-KO) mice, which develop hepatic steatosis, NASH, and HCC in the presence of obesity and insulin resistance [80]. Anagliptin treatment resulted in the suppression of hepatic crown-like structure formation and CD11c^+^ proinflammatory macrophage aggregation around dead hepatocytes, which were found in MC4R-KO mice fed a high-fat diet, suggesting a reduction in hepatic macrophage activation by anagliptin. Consequently, anagliptin attenuated hepatic inflammation, fibrosis, and carcinogenesis in MC4R-KO mice fed a high-fat diet. Regarding the molecular mechanism underlying macrophage inactivation by anagliptin, glucagon-like peptide-1 (GLP-1) was found to inhibit the upregulation of proinflammatory and fibrogenic genes in macrophages (Figure 3). DPP4 inhibitors increase the level of circulating GLP-1 to enhance insulin secretion [81]. Interestingly, anagliptin did not suppress the mRNA expression of inflammation-related genes in adipose tissue, even though these genes were upregulated in the adipose tissue of MC4R-KO mice fed a high-fat diet.

Dysregulation of M1/M2 polarization of macrophages is currently recognized to be a central mechanism underlying the pathogenesis of obesity and comorbidities such as insulin resistance and NAFLD [82,83]. M1 macrophage deletion increases sensitivity to insulin in obese mice [84,85], and a reduction in M2 macrophages predisposes lean mice to insulin resistance [86]. Therefore, inhibition of M1 polarization and/or alternative activation of M2 polarization of macrophages by DPP4 inhibitors may protect against exacerbation of inflammation and attenuate the progression to steatohepatitis and HCC.

#### 4.2.2. Metabolic Reprogramming

Aside from their inhibitory effects on insulin resistance and macrophage activation in NASH, DPP4 inhibitors have been reported to suppress the progression of NASH-related HCC through downregulation of nucleotide production [87]. Pentose phosphate pathway activity results in the production of ribose 5-phosphate, which is used in the synthesis of purine nucleotides and nucleic acids [88]. In the above-referenced study, the production of ribose 5-phosphate was increased, but purine nucleotide production was downregulated by DPP4 inhibitor treatment, suggesting that the DPP4 inhibitor suppressed the final enzymatic reaction in purine nucleotide synthesis. The transcription factor, nuclear factor erythroid 2-related factor 2 (Nrf2), activates purine nucleotide synthesis through enhancement of pentose phosphate pathway activity in proliferating cells under the sustained activation of phosphoinositide 3-kinase (PI3K)-Akt signaling [89]. The expression of Nrf2 in HCC tissue was suppressed by DPP4 inhibitor treatment [87], suggesting that Nrf2 downregulation resulted in the suppressed synthesis of purine nucleotides. However, the mechanisms by which DPP4 inhibitors suppress the expression of Nrf2 remains unclear.

Thus, DPP4 is important as a therapeutic target in HCC that arises from NASH and liver cirrhosis, but it remains unknown whether it is true for HCC that arises de novo in the otherwise healthy liver such as fibrolamellar HCC. Interestingly, *DNAJB1-PRKACA* fusion transcript was reported to induce liver innate immune inflammation in a zebrafish model of fibrolamellar HCC [90], suggesting that pharmacological inhibition of TNFα or caspase-1 activity might be targets to treat inflammation and progression in patients with fibrolamellar HCC. In this regard, it seems to be worth investigating whether DPP4 might be a therapeutic target in other clinical types of HCC.

## 5. Inhibition of CD26/DPP4 Non-Enzymatic Activity

### Ferroptosis

Ferroptosis is a new program of iron-dependent regulated cell death that is initiated through excessive peroxidation of polyunsaturated fatty acids and differs from apoptosis, necrosis, necroptosis, pyroptosis, and autophagy [91]. Ferroptosis has been found to occur in HCC [92]. Tumor suppressor p53 (TP53) was demonstrated to promote ferroptosis by transcriptionally repressing the expression of SLC7A11 (a specific light chain subunit of the cystine/glutamate antiporter) [93,94,95,96]. In contrast, a recent study showed that TP53 limits ferroptosis by blocking DPP4 activity in a transcription-independent manner [97]. TP53 promotes the subcellular redistribution of DPP4 toward a nuclear, enzymatically inactive pool. DPP4 typically functions as a serine protease in the plasma membrane but can act as a transcription cofactor in the nucleus [98]. TP53-mediated expression of SLC7A11, which contributes to ferroptosis resistance, is regulated by the DPP4-TP53 complex in the nucleus. In contrast, loss of TP53 prevents nuclear accumulation of DPP4 and thus facilitates plasma membrane-associated DPP-4-dependent lipid peroxidation via interaction with NADPH oxidase 1 (NOX1). This event finally results in ferroptosis, but the enzymatic activity of DPP4 is not essential for ferroptosis induction. These results suggest that DPP4 interacts with NOX1 in a non-enzymatic manner. Interestingly, DPP4 inhibitors (vildagliptin, alogliptin, and linagliptin) have been shown to completely block this plasma membrane-associated DPP-4-dependent ferroptosis in the context of TP53 loss [97]. In general, cell death is a tumor-promoting mechanism mediated by increased compensatory regeneration, fibrogenesis, and inflammation [99]. However, we should carefully distinguish between cell death in nontransformed cells and cell death in transformed cells because these programs have opposite functional consequences. Ferroptosis is no exception in this regard; that is, it mediates tumor promotion in nontransformed cells and tumor regression in transformed cells.

Thus, DPP4 inhibitors can target both the enzymatic and non-enzymatic functions of DPP4 [100,101,102,103,104]. DPP4 also has other non-enzymatic functions that are unrelated to its dipeptidase activity, in which it interacts with different partners and sustains tumor growth, invasion, and metastasis [17]. In addition to forming heterodimers with fibroblast activation protein alpha [105], CD26/DPP4 has been reported to associate with plasminogen 2 [106], adenosine deaminase [107,108], CD45 [109,110], CXCR4 [111], and mannose-6-phosphate/insulin-like growth factor II receptor [112] with its beta propeller domain containing binding sites for several proteins. Compared to enzymatic activity-dependent mechanisms of action of DPP4 inhibitors, enzymatic activity-independent mechanisms of action of DPP4 inhibitors have not been fully clarified. Further studies are required to clarify this issue.

## 6. Concluding Remarks

In this review, we focused on the biological roles of CD26/DPP4 in the development and progression of HCC and discussed the potential of CD26/DPP4 as a therapeutic target in HCC. In particular, immunomodulation by DPP4 inhibitors appears to be important in the development of therapeutic options for HCC because activation of lymphocyte trafficking by DPP4 inhibitors targeting CXCL10 [33] and the presence of truncated CXCL10 in serum [47,48,49,50] have already been confirmed in patients with HCC. These results shed light on targeted immunotherapy for advanced HCC, considering the lack of therapeutic agents that can promote lymphocyte trafficking to the cancer microenvironment.

Obesity and insulin resistance are closely related to hepatocarcinogenesis. Dysregulation of M1/M2 polarization of macrophages is currently recognized to be a central mechanism underlying the pathogenesis of obesity and comorbidities such as insulin resistance and NAFLD [82,83]. Because DPP4 inhibitors have been demonstrated to suppress M1 polarization and/or alternative activation of M2 polarization of macrophages in the liver and adipose tissue in mouse models, these results suggest the potential of DPP4 inhibitors as metabolic modulators for preventing hepatocarcinogenesis. Thus, CD26/DPP4 may be a critical and promising target for the prevention of HCC development and progression. However, most results have been obtained only in experimental animal models, and some issues, such as whether DPP4 has an inhibitory or promotive effect on cancer metastasis, are controversial. Therefore, further studies are required for the clinical translation of DPP4 inhibitors as anticancer agents for HCC.

## Figures and Tables

**Figure 1 cancers-14-00454-f001:**
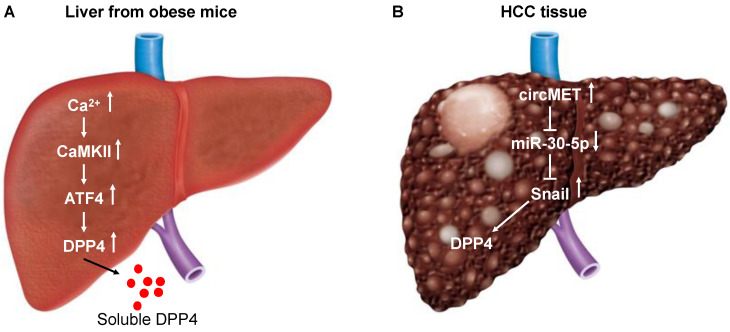
Schematic diagram depicting the induction of CD26/DPP4 expression in the liver of obese mice (**A**) and in HCC tissue (**B**). (**A**) In obese mice and humans, a pathway in hepatocytes involving Ca^2+^-calmodulin-dependent protein kinase II (CaMKII) regulates activating transcription factor 4 (ATF4). Activation of ATF4 leads to induction of dipeptidyl peptidase (DPP4) and secretion of soluble DPP4. (**B**) MET proto-oncogene receptor tyrosine kinase (*MET*) derived circular RNA (circRNA) (circMET) is the most overexpressed in HCC tissues compared to non-tumor liver tissues among the circRNAs derived from the chromosome 7q21–7q31 region. CircMET acts as a sponge for microRNA (miR)-30-5p, which downregulates Snail. Snail upregulates DPP4 expression by interacting with the enhancer element (region between 140,810 and 96,828 base pairs upstream of the DPP4 translation start site) of DPP4.

**Figure 2 cancers-14-00454-f002:**
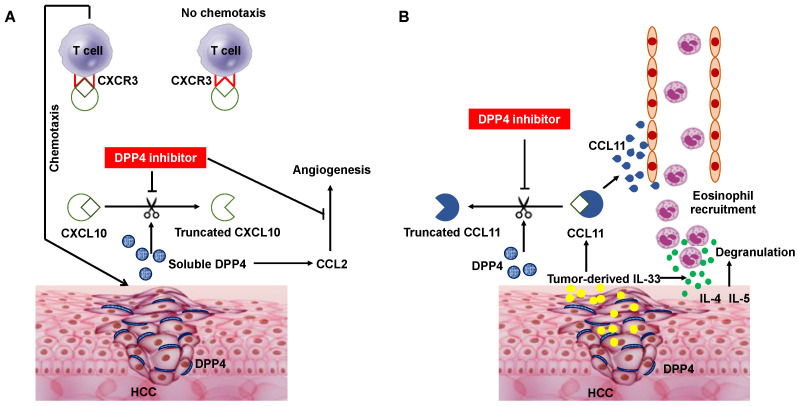
Schematic diagram depicting DPP4 inhibitor-mediated T cell chemotaxis activation and angiogenesis inhibition (**A**), and eosinophil recruitment (**B**). (**A**) DPP4 expression and the soluble DPP4 concentrations are increased in tumor tissue and serum in HCC patients, respectively. Inhibition of DPP4 enzymatic activity enhances anti-tumor effects by preventing the biologically active form of CXCL10 from being truncated by active DPP4 and increasing lymphocyte trafficking into the tumor. DPP4 inhibitor treatment also inhibits angiogenesis by downregulating chemokine ligand 2 (CCL2) expression induced by soluble DPP4. (**B**) CCL 11, an eosinophil-selective chemokine in the eotaxin subfamily, is induced by tumor-derived IL-33, which also stimulates eosinophil degranulation and recruits eosinophils toward HCC tumors. CCL11 is cleaved by DPP4 into a truncated form (CCL11_3–74_) with diminished chemoattractant properties. Consequently, DPP4 inhibitor treatment activates anti-tumor immunity through enhancement of eosinophil migration, independent of lymphocyte trafficking. CXCR3, CXC chemokine receptors 3.

**Figure 3 cancers-14-00454-f003:**
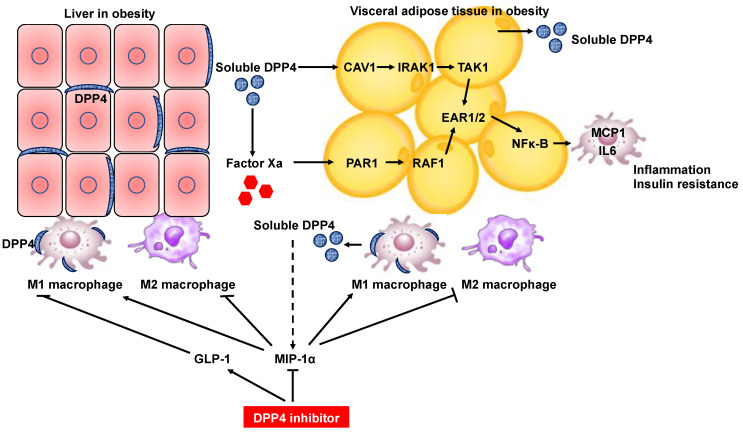
Schematic diagram depicting DPP4 inhibitor-mediated inhibition of inflammation in the liver and adipose tissues through amelioration of dysregulated M1/M2 polarization of macrophages. Hepatocyte DPP4 expression in the context of obesity results in secretion of soluble DPP4, which interacts with factor Xa. Soluble DPP4 and factor Xa activate two separate pathways (the CAV1-IRAK1-TAK1 pathway for DPP4 and the PAR2-RAF1 pathway for factor Xa), which synergistically stimulate ERK1/2 and NF-κB to induce MCP1 and IL-6 expression in adipose tissue macrophages. Silencing of DPP4 expression in hepatocytes but not treatment with DPP4 inhibitors suppress inflammation of VAT. Soluble DPP4 is also released from adipose tissue as an adipokine in obese individuals. DPP4 is predominantly expressed in M1-polarized macrophages compared with adipocytes in white adipose tissue of diet-induced obese mice. DPP4 inhibitor treatment induces a reciprocal decrease in M1 macrophages and an increase in M2 macrophages in white adipose tissue and liver tissue of diet-induced obese mice. MIP-1α becomes the most efficient monocyte chemoattractant after cleavage by DPP4 and is potentially a substrate of DPP4 for the regulation of macrophage polarization in the obese state. DPP4 inhibitor treatment also suppresses the upregulation of proinflammatory and fibrogenic genes in hepatic macrophages through enhancement of the circulating GLP-1 level in genetically obese NASH mice. CAV1, caveolin-1; IRAK1, interleukin-1 receptor-associated kinase; TAK1, TGF-β activated kinase 1; PAR2, protease-activated receptor 2; ERK, extracellular signal-regulated kinase; NF-κB, nuclear factor kappa-light-chain-enhancer of activated B cells; MCP1, monocyte chemokine protein 1; GLP-1, glucagon-like peptide-1.

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
