# Peer review of "CD26/DPP4 as a Therapeutic Target in Nonalcoholic Steatohepatitis Associated Hepatocellular Carcinoma"

_cancers, 2022, doi:10.3390/cancers14020454_

Round 1

Reviewer 1 Report

The review is complete and informative for the readers as concern the topic. 

However, it should be interesting to report some notes about the clinical perspectives of DPP4 inhibitors in prevention and treatment of hepatocellular carcinoma.

Moderate English editing (some orthographic errors are present in the text) is required. 

Author Response

We thank the reviewer 1 for his/her careful reading of the manuscript and appreciate their constructive comments. We have revised the manuscript to address each concerns raised by reviewers. Revised parts are highlighted in yellow.

According to the reviewer’s comments, we have added the report which showed the clinical perspectives of DPP4 inhibitors in prevention and treatment of HCC (lines 182-185 on page 9), and corrected some orthographic errors.

Reviewer 2 Report

This is an interesting review focused on CD26/DPP4 as a therapeutic target in hepatocellular carcinoma (HCC). In addition, the authors also concluded potential role of CD26/DPP4 in the development and progression of HCC.

The need of this review rises from some general aspect on current issues of non-alcoholic hepatitis (NASH) associated HCC.

The author has also presented results from recent studies about representative small molecules.

This review is well written and comprehensive.

I have a couple of recommendations.

The title would be changes as “CD26/DPP4 as a therapeutic target in non-alcoholic hepatitis associated hepatocellular carcinoma”.

It would be necessary to add a mechanism of low efficacy of PD-L1-based therapy treatment in NASH-associated HCC and a possible role of CD26/DPP4 inhibition.

Author Response

We thank the reviewer 2 for his/her careful reading of the manuscript and appreciate their constructive comments. We have revised the manuscript to address each concerns raised by reviewers. Revised parts are highlighted in yellow.

According to the reviewer’s comments, we have changed the title as “CD26/DPP4 as a therapeutic target in nonalcoholic steatohepatitis associated hepatocellular carcinoma”.

According to the reviewer’s comments, we have added mechanisms of low efficacy of PD-L1-based therapy in NASH-associated HCC and a possible role of CD26/DPP4 inhibition (lines 212-217 on pages 10 and 11).

Reviewer 3 Report

The authors of this manuscript have presented a compelling case for the role of CD26/DPP4 as a molecule with a substantial role in the pathogenesis of hepatocellular carcinoma. The manuscript presents previously published data from reliable, peer reviewed sources.

Perhaps my biggest question surrounds the role of CD26/DPP4 in what can be termed as "atypical" HCC, specifically that of HCC that arises de novo in an otherwise healthy liver as well as fibrolamellar HCC.

With regard to de novo HCC, has there been any investigation in human liver tumors whether the over expression of DPP4 and its cleaved subunit is true as well. Could the same be asked about FLHCC. Although the driver event for FLHCC via the DNAJB1-PRKACA fusion protein has been well published, some investigators speculate that FLHCC has an inflammatory phenotype. It would be interesting to see if such data exists.

In this regard, perhaps the authors could comment and broaden their hypothesis that DPP4 is important not only for HCC that arises from NAFLD/NASH and cirrhosis, but perhaps other clinical "types" of HCC as well.

Additionally, can the authors please elaborate further on clinical assessment of CD26 as follows:

  1. Have any studies looked at the IHC expression of DPP4 specifically in human HCC? It would appear that the citations provider have gone no further than investigation in human PDXs, rather than clinical specimens.
  2. Have any investigators looked at the genomic and histologic expression of DPP4 in HCC metastases? I think such investigations would also be interesting from a possible therapeutic standpoint as metastatic HCC is the ultimate treatment challenge.

Author Response

We thank the reviewer 3 for his/her careful reading of the manuscript and appreciate their constructive comments. We have revised the manuscript to address each concerns raised by reviewers. Revised parts are highlighted in yellow.

According to the reviewer’s comments, we have added the paragraph which describes the role of liver innate immune inflammation in fibrolamellar HCC and the potential of DPP4 as a therapeutic target in clinical types of HCC that arises de novo in an otherwise healthy liver (lines 360-367 on page 17).

We studied the immunohistochemical expression of CD26 in resected HCC specimens obtained from 41 patients. We have reported that HCC tumors with high CD26 expression showed higher alpha-fetoprotein levels, more advanced tumor stages, poorer histologic differentiation, higher tumor cell infiltration into the tumor capsule, and greater cell proliferation than those with lower CD26 expression, and that the cumulative recurrence rate of HCC was higher and the overall survival time was shorter in HCC patients with high CD26 expression [11] (lines 81-87 on pages 4 and 5).

Unfortunately, we could not find the reports that showed the genomic and/or histologic expression of DPP4 in HCC metastases.